# Prevalence of Multidrug-Resistant and ESBL-Producing Bacterial Pathogens in Patients with Chronic Wound Infections and Spinal Cord Injury Admitted to a Tertiary Care Rehabilitation Hospital

**DOI:** 10.3390/antibiotics12111587

**Published:** 2023-11-02

**Authors:** Reem Binsuwaidan, Mohammad Aatif Khan, Raghad H. Alzahrani, Aljoharah M. Aldusaymani, Noura M. Almallouhi, Alhanouf S. Alsabti, Sajjad Ali, Omar Sufyan Khan, Amira M. Youssef, Lina I. Alnajjar

**Affiliations:** 1Department of Pharmaceutical Sciences, College of Pharmacy, Princess Nourah bint Abdulrahman University, P.O. Box 84428, Riyadh 11671, Saudi Arabia; rabinsuwaidan@pnu.edu.sa; 2Microbiology Laboratory, Department of Pathology and Laboratory Medicine, King Abdullah Bin Abdul Aziz University Hospital, Princess Nourah bint Abdulrahman University, P.O. Box 84428, Riyadh 11671, Saudi Arabia; mohaatif.khan@gmail.com; 3College of Pharmacy, Princess Nourah bint Abdulrahman University, P.O. Box 84428, Riyadh 11671, Saudi Arabia; raghad.hzzz@gmail.com (R.H.A.); aljwharaald@gmail.com (A.M.A.); nouraalmallouhi@gmail.com (N.M.A.); sabti.alhanouf@gmail.com (A.S.A.); 4Infectious Diseases, Medical Affairs Department, Sultan Bin Abdulaziz Humanitarian City, P.O. Box 64399, Riyadh 11536, Saudi Arabia; sajjad_siut@yahoo.com (S.A.); okhan@sbahc.org.sa (O.S.K.); 5Research and Scientific Center, Sultan Bin Abdulaziz Humanitarian City, P.O. Box 64399, Riyadh 11536, Saudi Arabia; ayouseff@sbahc.org.sa; 6Department of Pharmacy Practice, College of Pharmacy, Princess Nourah bint Abdulrahman University, P.O. Box 84428, Riyadh 11671, Saudi Arabia

**Keywords:** pressure ulcer, antimicrobial resistance, spinal cord injury

## Abstract

A pressure ulcer is defined as a skin lesion of ischemic origin, a condition that contributes to morbidity and mortality in patients with spinal cord injuries. The most common complication of ulcers is a bacterial infection. Antimicrobial therapy should be selected with caution for spinal cord injury patients since they have a high risk of developing multidrug-resistant (MDR) infections. The aim of this study was to determine the prevalence of different bacterial pathogens in patients with pressure ulcers admitted with spinal cord injuries. This was a retrospective single-center study that included adult patients aged 18 years and above, admitted with chronic pressure wounds after a spinal cord injury requiring hospitalization between 2015 and 2021. A total of 203 spinal cord injury patients with pressure ulcers were included in the study. Ulcers were commonly infected by *Staphylococcus aureus*, *Pseudomonas aeruginosa*, and *Escherichia coli*, and they were mostly located in the sacral and gluteal areas. More than half of the bacteria isolated from patients were sensitive to commonly tested antibiotics, while 10% were either MDR- or pan-drug-resistant organisms. Of the MDR bacterial isolates, 25.61% were methicillin-resistant *S. aureus*, and 17.73% were extended-spectrum beta-lactamase *Enterobacteriaceae.* The most prevalent bacteria in pressure ulcers of spinal cord injury patients were *S. aureus*. Other antibiotic-resistant organisms were also isolated from the wounds.

## 1. Introduction

Pressure ulcers are a serious complication for spinal cord injury patients, which significantly increases morbidity and mortality among this population [1,2]. The National Pressure Ulcer Advisory Panel defines a pressure ulcer as a skin lesion of ischemic origin related to the compression of soft tissues between a hard surface and a bony prominence [1]. Pressure ulcers form due to continuous pressure over a bony prominence, which results in shearing and/or ischemia of the overlying skin, leading to tissue breakdown [3]. A recent study showed that spinal cord injury patients are significantly more likely to develop ulcers, with incidences ranging from 10.2% to 30% [1]. Another study suggested that 25–50% of spinal cord injury patients require therapy for pressure ulcers [4]. The management of pressure ulcers is a huge medical burden associated with high costs of patient care [3]. Comprehensive management of pressure ulcers in spinal cord injury patients is needed to protect the patient’s physical, psychological, and social wellbeing, as well as their overall quality of life, especially in geriatric populations, which have been shown to exhibit more aggressive forms of pressure ulcers [1,3].

Complications arising from pressure ulcers are associated with significantly increased rates of morbidity and mortality [3]. The most common complications are bacterial infections. Infected pressure ulcers result in prolonged patient hospitalization, which increases health-care costs, in addition to antimicrobial-resistant infections that may cause further morbidity and lead to additional treatment costs arising from infection control measures implemented to avoid patient-to-patient transmission [5]. Pressure ulcers can affect different sites of the body. The most common sites of pressure ulcers in spinal cord injury patients are the ischium (28%), the sacrum (17–27%), the trochanter (12–19%), and the heel (9–18%) [6]. With regard to bacteria affecting pressure ulcers, one study showed that *Staphylococcus aureus*, *Proteus mirabilis*, *Pseudomonas aeruginosa*, and *Enterococcus faecalis* are among the most common bacteria found in pressure ulcers [3]. In another study, pressure ulcers involving *S. aureus*, Gram-negative bacilli, or both were found in 77% of patients [7]. The most important problem in managing wound infections is bacterial resistance, which has been observed with *S. aureus* and coagulase-negative *Staphylococci* among Gram-positive species, and with *Escherichia coli*, *Klebsiella pneumonia,* and *P. aeruginosa* among Gram-negative species [5]. Chronic wounds, such as pressure ulcers, are often polymicrobial infections, containing multiple bacterial pathogens that are often more virulent and damaging to the host [8]. Additionally, infected pressure ulcers may result in soft tissue and bone infections, such as cellulitis, abscesses, bursitis, and osteomyelitis of the bone underlying the wound bed [3]. Notably, invasive wound infections may lead to life-threatening sepsis and septic shock [1].

Currently, there are no clinical trials or comparative studies of antimicrobial therapy, the course of treatment, or the best route of administration. The Infectious Disease Society of America (IDSA) developed guidelines for skin and soft tissue infections (SSTIs) [9]. The current guidelines recommend treatment for SSTIs, such as cutaneous abscesses, furuncles, carbuncles, and others. Based on research, the guidelines indicate that skin ulceration and cellulitis may be caused by Group A *Streptococci*, *S. aureus* and occasionally methicillin-resistant *S. aureus* (MRSA). Empiric treatment of uncomplicated cellulitis and skin ulceration should include antibiotics that are effective against *Streptococci* and *Staphylococci*, such as penicillin, amoxicillin, amoxicillin-clavulanate, dicloxacillin, cephalexin, and clindamycin.

Antimicrobial therapy should be considered with caution for patients with spinal cord injuries since they may have already been exposed to previous antimicrobial treatments and have a high rate of acquiring multidrug-resistant (MDR) organism infections [1]. In addition, antimicrobial therapy should be as short in duration as possible, based on the microbiology culture and susceptibility results, and must have suitable bioavailability.

To effectively prescribe antimicrobial therapy, the early symptoms of wound infections, causative pathogens, and their prevailing susceptibility patterns must be elucidated [5]. Although antibiotics play an important role in wound care and healing, prescribing random antibiotics may produce unintended consequences, such as the acquisition of MDR organisms, making wound healing more difficult due to limited blood flow and penetration of antimicrobials into the affected site [10]. In 2013, MRSA was found in 38.6% of people with a spinal cord injury within 48 h of admission to an acute care facility [11]. Therefore, topical and narrow-spectrum antibiotics are better for use unless clinical symptoms of infection progress; in this case, systemic antibiotic therapy should be considered. The European Pressure Ulcer Advisory Panel issued guidelines on the treatment of pressure ulcers, recommending that systemic antibiotics should be avoided for pressure ulcers with only clinical symptoms of local infection [12]. Theoretically, local/topical antibiotics have several advantages; they protect against systemic antimicrobial adverse effects, such as altering and diminishing the protective gut/vaginal flora, minimizing drug interactions, preventing systemic toxicity, and, importantly, reducing the risk of bacterial resistance [1]. Compared to non-antibiotic topical options, some moderate- and low-quality evidence suggests that povidone–iodine treatment results in fewer ulcers and short-term wound healing [13]. In addition, there is some evidence of a higher probability of healing ulcers when treated with non-antimicrobial alternate dressings than when treated with povidone–iodine dressings [7].

In general, MDR organism isolations in soft tissue samples are not well understood, which has become a major medical problem in hospitalized patients since MDR treatment options are limited and increase health-care costs [5]. Antimicrobial usage has become a common clinical practice for preventing and managing infections, but approximately 20% of antibiotics are inappropriately prescribed [14]. Most approaches for people with spinal cord injuries with pressure ulcers are based on experience and are not evidence-based, which exacerbates clinical outcomes following MDR infection [1]. Clinical trials aimed at addressing MDR infection in pressure ulcers are relatively small, clinically heterogeneous, generally short in duration, and have a high or unclear risk of bias. As a result, more randomized clinical studies with larger sample sizes are needed to understand the role of MDR organisms and their susceptibility to antibiotics [7].

The main aims of this study were as follows:To evaluate the prevalence of different bacterial pathogens in patients diagnosed with spinal cord injuries and pressure ulcers.To determine the prevalence of MDR and extended-spectrum beta-lactamase (ESBL)-producing Gram-negative bacteria isolated from chronic wounds.To investigate how frequently antibiotics are prescribed in patients with chronic wound infections.

We performed a retrospective data review of patients with spinal cord injuries who were admitted with infected pressure ulcers during a specified period. We analyzed the data to determine the different bacterial pathogens isolated from pressure wounds and the antibiotic selection criteria for treating such infections.

## 2. Results

We analyzed data from the hospital records of 203 patients. Baseline characteristics are summarized in Table 1. The majority of patients in this study were male (91.62%). The study included three age categories: 18–39 years (78.8%), 40–60 years (13.8%), and >60 years (7.4%). Traumatic spinal cord injury was more prevalent (90.15%) than non-traumatic, and 33% of patients were immobile, while 64.5% had limited mobility. Most patients had a single ulcer (61.08%), with only one bacterial species isolated from the ulcer (48.27%).

The characteristics of the pressure ulcers are shown in Table 2. Ulcer locations varied between patients: half of the ulcers were located in the sacral region, and 29% were located in the gluteal region. Most ulcers were infected (80%), and a few were colonized.

The patients showed different stages of ulcers based on clinical examination according to the National Pressure Ulcer Advisory Panel Pressure Injury Staging System [15], as follows: stage 2, 5.4%; stage 3, 24.63%; stage 4, 59.11%; and 10.83% of patients’ files did not indicate ulcer stage.

We analyzed the relationship between the site of infection and the isolated bacteria. The most commonly isolated organism from ulcers was *S. aureus*. Sacral infections were mainly due to *S. aureus* (28%), *P. mirabilis* (21%), and *P. aeruginosa* (16%). Gluteal infections were mostly caused by *S. aureus* (34%). The other ulcer locations reported in only 16 or fewer patients were infected by a variety of organisms, as illustrated in Figure 1.

The different antibiotics used to treat ulcers are shown in Table 3, which illustrates the number of antibiotic prescriptions for pressure ulcers. Most patients were treated with the application of a wound dressing, which was applied at the site of the ulcer. The majority (54.67%) received a non-antimicrobial dressing, followed by patients who received an antimicrobial dressing (39.90%). Approximately 10% of infected ulcers were treated with topical antibiotics, with mupirocin being the most commonly prescribed topical antibiotic. Fewer patients were treated with an oral antibiotic (15.27%), which included amoxicillin/clavulanic acid, clindamycin, and quinolones. Parenteral antibiotics (10.83%) were administered to some patients, with meropenem being prescribed most frequently, followed by ceftriaxone. A small percentage of patients (5.41%) did not receive any type of treatment for ulcers.

Table 4 shows the antibiotic susceptibility patterns of bacteria isolated from ulcers. The most common bacteria isolated from pressure ulcers were *S. aureus*, followed by *P. aeruginosa*, *E. coli*, and *K. pneumoniae*. Most *S. aureus* isolates were resistant to oxacillin, which classifies the isolate as MRSA. Resistance to ciprofloxacin, trimethoprim, and clindamycin was also common in *S. aureus*. Most *S. aureus* isolates were generally sensitive to clindamycin, mupirocin, trimethoprim, and linezolid. Susceptibility data for *S. aureus* to vancomycin were not available. By contrast, Gram-negative bacteria, such as *P. aeruginosa* isolates, were mostly susceptible to ceftazidime, ciprofloxacin, gentamycin, and meropenem. *E. coli* isolates were mostly resistant to ampicillin and susceptible to cephalosporins, ceftazidime, ceftriaxone, and cefepime, as well as gentamicin and trimethoprim. *K. pneumonia* isolates were few and sensitive to aminoglycosides and cefepime.

The susceptibility pattern of MDR is shown in Table 5. Although most (63.05%) isolated bacteria were sensitive to antibiotics, 7.88% were considered to be MDR and 2.46% were defined as pan-drug-resistant (PDR). A quarter of the isolates (25.61%) were MRSA, and ESBL organisms were isolated from ulcers in some patients (17.73%).

## 3. Discussion

Spinal cord injury patients have a high risk of developing pressure ulcers and require prolonged hospital treatment with a longer length of stay. Approximately 38% of spinal cord injury patients develop at least one pressure ulcer during the period of their hospitalization and rehabilitation [16]. Pressure ulcer infection is a serious complication that involves poor blood flow at the affected site in some patients, delayed wound healing, and a risk of bacterial colonization, which significantly increases the morbidity and mortality of this population [1,2]. The colonization of ulcers with MDR and PDR bacterial pathogens represents an emerging problem and can complicate clinical outcomes for spinal cord injury patients and rehabilitation centers [17]. Therefore, this study examined the prevalence of different organisms in patients with pressure ulcers who were admitted with spinal cord injuries, in addition to determining the prevalence of MDR organisms and their antibiotic susceptibility patterns among spinal cord injury patients with chronic wounds.

In this study, most ulcers were infected (80%), and a few were colonized, which was defined as the isolation of bacterial pathogens from a wound site without the presence of inflammatory markers, such as PMNs. These results support previous findings by Dinh, A. et al. [1], demonstrating that the majority of ulcers in patients with spinal cord injuries were infected with bacterial pathogens. *S. aureus* was the most commonly isolated organism from ulcers. Sacral ulcer infections were mainly infected by *S. aureus* (28%), *P. mirabilis* (21%), and *P. aeruginosa* (16%). By contrast, gluteal infections were mostly caused by *S. aureus* (34%). Other ulcer locations were infected by *S. aureus*, *P. aeruginosa*, *P. mirabilis*, *K. pneumoniae*, and *E. coli*. This is largely due to the presence of *Staphylococci*, which colonizes skin surfaces in the groin and at other sites of the body. The presence of an ulcer increases the likelihood that Gram-positive bacteria will contribute to an infection. By contrast, Gram-negative bacteria in the *Enterobacteriaceae* family, such as *K. pneumonia* and *E. coli*, are likely acquired from the enteric flora of the patient’s feces, which may contaminate wounds in the lower abdomen. Alternatively, other Gram-negative pathogens, such as the *Pseudomonas* species, may result in health care-acquired infections following exposure to the hospital environment. Our findings are consistent with previous studies that identified *S. aureus* and *Enterobacteriaceae* as the most common bacterial species isolated from wounds [10,18]. Taken together, our results suggest that the predominant organisms identified in chronic ulcers are largely similar across various studies with different study populations.

The main contributors to antimicrobial drug resistance are the lack of evidence-based practice, the overuse of antibiotics, and the inefficient prevention and control practices in health-care settings; therefore, determining the prevalence of drug-resistant organisms is crucial in the selection of antibiotics that target specific pathogens and minimize unnecessary use of broad-spectrum antibiotics [1]. An understanding of the differences in colonization versus infection of chronic wounds and pressure sores is useful in clinical decision-making when deciding whether to treat such wounds with antibiotics. Patients with pressure sores are often likely to be bedridden for longer periods, especially following a spinal cord injury. In older patients, comorbidities, such as diabetes and obesity, are major contributing factors in poor wound healing due to ischemia, necrosis, and devitalized tissue at the wound site. Consequently, the immune response and healing may be significantly impaired and could negatively impact host defenses. In such a milieu, bacteria are likely to colonize these sites without initiating an immune response, which may be dampened [19]. A Gram stain obtained from a wound swab is valuable in such circumstances, as it may reveal whether PMNs are present at the site. Moderate to many PMNs reported in a Gram stain with abundant bacteria likely correlate with an infectious process and possibly warrant antimicrobial therapy. In contrast, the isolation of scant bacteria without a significant number of PMNs could reflect colonization and does not require therapy. Studies have shown that treating colonization leads to a transient clearance of bacteria, which are likely to recolonize chronic wounds over time. Importantly, repeated antibiotics may select out resistant bacteria, such as MRSA and ESBL, which, once colonized, can be very difficult to eradicate [20].

In this study, we found the antibiotic susceptibility patterns of bacteria isolated from pressure ulcers among spinal cord injury patients: 63.05% of the bacterial isolates were sensitive to the antibiotics that were tested, 7.88% were MDR, and only 2.46% of the isolates were PDR that were resistant to all the antibiotics tested in the laboratory. The most frequently isolated resistant bacteria were MRSA (25.61%) and ESBL (17.73%). Other studies have reported higher rates of resistance in bacteria isolated from wound infections (67.1%) [17] and approximately 39% of MRSA from ulcers in spinal cord injury patients [11].

The majority of the patients in our study were treated with non-antimicrobial dressings (54.67%), which is the standard practice at the rehabilitation hospital. However, for some symptomatic infections of wounds, oral antibiotics were administered for treatment. Before antimicrobial therapy, it is important to identify the prevalence of bacterial infections and their antibiotic susceptibility patterns so that the right antibiotic can be selected to combat the infection, accelerate wound healing, and improve recovery [5,14].

We would like to emphasize the value of physicians reviewing hospital antibiograms before making clinical decisions about infectious diseases. The hospital antibiogram is cumulative antibiotic susceptibility data that are prepared, at least annually, by the microbiology laboratory [21,22]. The antibiogram provides percentage susceptibility data from the previous year for the most frequently isolated bacterial pathogens. The antibiogram can, therefore, facilitate the selection of antibiotics for empiric therapy before culture and susceptibility results are available from the microbiology laboratory. By reviewing the antibiogram, a physician can potentially assess which bacterial strains were common in the hospital in the previous year and make evidence-based therapeutic decisions.

A potential limitation to this study is the small sample size; only 203 patients were included from a single-center study. However, the data collection method was precise, and the data access was available during and after the data collection period, which affected the accuracy of the results. An unexpected confounding factor that may call into question the relevance of the study is the baseline characteristic that 91.62% of the included patients were male, which was the majority of the sample size. However, this is likely due to the regional lifestyle choice that predominantly males drive cars, thus accounting for the higher number of trauma-related road traffic accidents. Additionally, the previous antibiotic pressure of the included patients was not reflected in this study; this point could have a strong relationship with the type of bacteria isolated.

## 4. Materials and Methods

### 4.1. Study Design and Setting

Our study is a retrospective single-center study that included adult patients aged 18 years and older. The study’s duration was six months for patients who were admitted with chronic pressure ulcers after a spinal cord injury and required hospitalization between November 2015 and May 2021 in Sultan bin Abdulaziz Humanitarian City, Riyadh, Saudi Arabia. Pediatric patients under 18 years and patients with any diagnosis other than a spinal cord injury were excluded from this study.

### 4.2. Data Collection

Our research data were collected from patients’ electronic medical files in the Sultan bin Abdulaziz Humanitarian City hospital information system. The collected data encompassed all patient information, including demographics, clinical presentation, and spinal cord injury details, which included the number of ulcers, their locations, and whether the wounds were infected or not.

The criteria used to differentiate between colonization and infection were based on a clinical assessment of the following signs or symptoms: abscess; cellulitis; purulent discharge; granulation tissue; unexpected pain/tenderness; and systemic signs, such as fever or the presence of inflammatory immune cells, such as polymorphonuclear (PMN) leukocytosis. Data on infected wounds included the identified bacterial pathogens and their antibiotic susceptibility results. The antibiotics used for treatment were also documented in a detailed manner, including their route of administration and how frequently the antibiotic was prescribed. MDR organisms are generally resistant to three or more classes of antibiotics, while PDR implies resistance to all antibiotics tested in the laboratory. Data on patients’ functional status, including physical and mental status, and bladder and bowel incontinence, were collected. Any comorbidities, dressing types, and other treatment options were also recorded.

### 4.3. Statistical Analysis

The sample size was calculated using Epi Info Version 7, under the assumption that the prevalence of ESBL was 19.9% and a 5% confidence level, as reported by Zowawi et al. [23]. The minimum required sample size at a 95% confidence level was 178 patients, which was rounded to 200 patients. The statistical analysis was completed accordingly. Descriptive statistics were used to determine bacterial prevalence and patterns of resistance. Descriptive statistics included counts and proportions for categorical variables. The data analysis was conducted using the Statistical Package for Social Sciences (SPSS) version 21 (SPSS Inc., Armonk, NY, USA).

## 5. Conclusions

This study revealed the characteristics of bacterial pathogens isolated from pressure ulcers in patients with spinal cord injuries. Understanding the prevalence of such bacteria in wounds and their antibiotic susceptibility patterns can inform an improved choice of targeted antibiotic therapy. Our findings will hopefully promote better use of available antibiotics, especially when deciding on empiric therapy. Further studies are needed to identify the most appropriate management approach for pressure ulcers to improve patients’ responses to treatments and reduce the length of hospital stays and health-care costs.

## Figures and Tables

**Figure 1 antibiotics-12-01587-f001:**
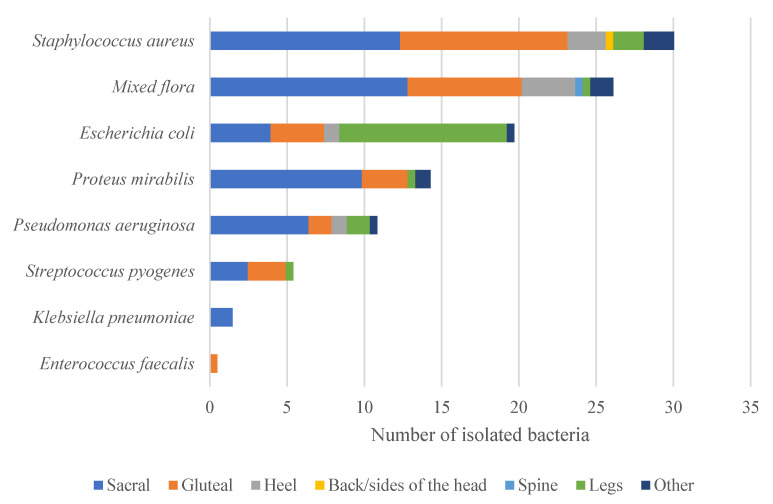
Number of isolated bacteria based on ulcer location.

**Table 1 antibiotics-12-01587-t001:** Baseline characteristics of patients.

Character		Number (*N*)	Frequency (%)
Total		203	100.00
Gender	Male	186	91.62
	Female	17	8.38
Age	18–39	160	78.88
	40–60	28	13.79
	>60	15	7.39
Hospitalization time (weeks)	≤4	27	13.30
	5–8	65	32.02
	9–12	72	35.47
	≥13	39	19.21
Trauma	Traumatic	183	90.15
	Non-traumatic	20	9.85
Mobility	Full mobility	5	2.47
	Limited mobility	131	64.53
	Immobile	67	33.00
Underlying condition	Hypertension	18	8.86
	Diabetes	25	12.31
	Dyslipidemia	5	2.46
	Organ failure	1	0.49
	Asthma	5	2.46
	Obesity	18	8.86
Number of ulcers	1	124	61.08
	2	63	31.04
	3	14	6.89
	Multiple	2	0.99
Number of bacterial isolates	1	98	48.27
	2	72	35.45
	3	14	6.89

**Table 2 antibiotics-12-01587-t002:** Pressure ulcer characteristics.

Character		Number of Ulcers (*N*)	Frequency (%)
Ulcer Location	Sacral	101	49.76
	Gluteal	59	29.08
	Heel	16	7.88
	Back or side of the head	1	0.49
	Spine	1	0.49
	Legs	13	6.40
	Other	12	5.90
Wound	Colonized	42	20.68
	Infected	161	79.31
Ulcer Stage	Level 2	11	5.41
	Level 3	50	24.63
	Level 4	120	59.11
	N/A	22	10.83

N/A: Not available.

**Table 3 antibiotics-12-01587-t003:** Frequency of antibiotic prescriptions for pressure ulcers.

Antibiotics		Prescription Frequency *N* (%)	Ulcer Stage			
			2	3	4	Not Known
Total		203 (100)	11 (5.41)	50 (24.63)	120 (59.11)	22 (10.83)
Topical	Fucidin topical cream	1 (0.49)	-	1 (2)	-	-
	Mupirocin ointment	8 (3.94)	-	4 (8)	4 (3.33)	-
	Silver sulphadiazine cream	1 (0.49)	-	1 (2)	-	-
	Triamcinolone, nystatin, neomycin, gramicidin	4 (1.97)	-	1 (2)	1 (0.83)	2 (9.09)
Oral	Amoxicillin/clavulanic acid	17 (8.37)	1 (9.09)	8 (16)	7 (5.83)	1 (4.54)
	Cefuroxime	2 (0.99)	-	-	2 (1.67)	-
	Ciprofloxacin	3 (1.48)	-	-	2 (1.67)	-
	Clindamycin	7 (3.45)	-	-	6 (5)	1 (4.54)
	Levofloxacin	1 (0.49)	-	1 (2)	-	-
	Rifampin	1 (0.49)	-	-	1 (0.83)	-
	Sulfamethoxazole, trimethoprim	1 (0.49)	-	-	1 (0.83)	-
Systemic	Amoxicillin, clavulanic acid	2 (0.99)	-	-	2 (1.67)	-
	Cefotaxime	1 (0.49)	-		1 (0.83)	-
	Ceftriaxone	5 (2.45)	-	4 (8)	1 (0.83)	-
	Ciprofloxacin	2 (0.99)	-	-	2 (1.67)	-
	Clindamycin	1 (0.49)	-	-	1 (0.83)	-
	Linezolid	1 (0.49)	-	-	1 (0.83)	-
	Meropenem	9 (4.43)	-	1 (2)	7 (5.83)	1 (4.54)
Dressing	Antimicrobial dressing	135 (66.50)	6 (54.55)	35 (70.0)	83 (69.17)	11 (50.00)
	Non-antimicrobial dressing	58 (28.57)	4 (36.36)	13 (26.0)	32 (26.67)	9 (40.90)
	Not treated	10 (4.93)	1 (9.09)	2 (4.0)	5 (4.17)	2 (9.09)

**Table 4 antibiotics-12-01587-t004:** Antibiotic resistance pattern of bacteria isolated from ulcers (number of isolates sensitive or resistant to each antibiotic).

Bacteria (*n*)	* S. aureus *	* P. aeruginosa *	* E. faecalis *	Coagulase-Negative *Staphylococci*	* E. coli *	* S. pyogenes *	* K. pneumoniae *
Ampicillin	-	-	1 (R)	-	13 (R), 7 (S)	7 (S)	3 (R)
Amikacin	-	-	-	-	6 (S)	-	1 (S)
Ceftazidime	-	1 (R), 17 (S), 1 (I)	-	-	4 (R), 2 (S)	-	2 (R)
Clindamycin	4 (R), 58 (S)	-	-	1 (S)	-	6 (R), 5 (S)	-
Ciprofloxacin	8 (R), 4 (S)	2 (R), 16 (S), 1 (I)	-	-	6 (R)	-	2 (R), 1 (I)
Ceftriaxone	-	-	-	-	4 (R), 2 (S)	11 (S)	2 (R)
Cefepime	-	1 (R), 16 (S), 1 (I)	-	-	4 (R), 2 (S)	-	2 (S)
Gentamicin	1 (R)	1 (R), 18 (S)	-	-	5 (R), I (S)	-	1 (R), 2 (S)
Linezolid	32 (S)	-	-	-	-	-	-
Meropenem	-	1 (R), 16 (S), 2 (I)	-	-	4 (S)	-	1 (S)
Mupirocin	28 (S)	-	-	-	-	-	-
Oxacillin	35 (R), 23 (S)	-	-	1 (S)	-	-	-
Rifampin	4 (I), 33 (S)	-	-	-	-	-	-
Trimethoprim	6 (R), 23 (S)	-	-	1 (S)	3 (R), 3 (S)	2 (S)	2 (R), 1 (S)
Tigecycline	2 (S)	7 (R)	-	-	5 (R)	3 (S)	1 (S)

R = resistant; S = sensitive; I = intermediate.

**Table 5 antibiotics-12-01587-t005:** Resistance patterns of isolated bacteria.

Pattern	*N*	%
Sensitive	128	63.05
MDR	16	7.88
PDR	5	2.46
MRSA	52	25.61
ESBL	36	17.73

MDR: Multidrug-resistant, PDR: Pan-drug-resistant, MRSA: Methicillin-resistant *S. aureus,* ESBL: Extended-spectrum beta-lactamase.

## Data Availability

The data presented in this study are available on request from the corresponding author. The data are not publicly available due to patients’ information privacy.

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
