# Peer review of "Prevalence of Multidrug-Resistant and ESBL-Producing Bacterial Pathogens in Patients with Chronic Wound Infections and Spinal Cord Injury Admitted to a Tertiary Care Rehabilitation Hospital"

_antibiotics, 2023, doi:10.3390/antibiotics12111587_

Round 1
Reviewer 1 Report
Comments and Suggestions for Authors
In general, I like the work that you did, and I appreciate your effort to understand the prevalence of different bacterial pathogens in patients with pressure ulcers admitted with spinal cord injury between 2015 and 2021 . Almost all collected data analysis were performed correctly. Nevertheless, I have one serious remark which in my opinion is necessary additional data analysis for better conclusions.
Here is my observation. In table 1, baseline characteristics of patients are summarized, however a better analysis for age (18-39; 40-60 and ˃60) and period (2015-2021) is necessary to understand the “ The main aim of this study, which is to evaluate the prevalence of different bacterial patho-gens in patients diagnosed with spinal cord injuries and pressure ulcers as well as to determine the prevalence of MDR and extended-spectrum beta-lactamase (ESBL) producing Gram-negative bacteria isolated from chronic wounds”, because hospitalization time could be important for bacterial colonization and infection.
In figure 1, “other bacteria”, in necessary add bacterial names.
Data from table 2 and 3, a deep analysis is required to understand the correlation between pressure ulcer characteristics and antibiotics prescription for isolated bacteria.
Author Response
Attached the review response.

Reviewer 2 Report
Comments and Suggestions for Authors
The article “Prevalence of Multi-Drug Resistant and ESBL Producing Bacterial Pathogens in Patients with Chronic Wound Infections and Spinal Cord Injury Admitted to a Tertiary Care Rehabilitation Hospital” investigates the prevalence of different bacterial pathogens in patients diagnosed with spinal cord injuries and pressure ulcers and their resistance profiling. Moreover, the study aims to evaluate the treatment regime of the pressure ulcer. The study showed that spinal cord injury ulcers are commonly infected by Staphylococcus aureus, Pseudomonas aeruginosa and Escherichia coli and majority of them are not multidrug resistant. The study is well designed and very interesting but lack statistical analysis to strength the claims made by the authors. The following minor points should be addressed to the improve the quality of manuscript.
Comments
The introduction section needs to be a bit precise down.
Line 133 The aims of the study should be enlisted as points.
Line 143 It would be interesting to identify the risk factor associated with pressure ulcer by statistical analyzing (p values) the data presented in table 1 and 2.
Line 175 Figure 1 please add x-axis legend and remove the figure border.
Figure 2 needs to be improved.
Line 132 one of the aims of the study was investigated “how frequently antibiotics are being misused in such patients with chronic wound infections” but no statistically significant evidence is provided.
Comments on the Quality of English LanguageThe materials and methods section needs to be revised to remove minor grammatical errors.
Author Response
Attached the review response.

Reviewer 3 Report
Comments and Suggestions for Authors
The aim of the study was to determine the prevalence of different bacterial pathogens in patients with pressure ulcers (PU) admitted with spinal cord injury. Authors analyzed the medical files of 203 patients with PU and characterized the microbial factor and antibiotic resistance. The authors summarized all results in tables and charts wich is clear for the readers. However in Table 1 "Table 1. Baseline characteristics of patients." the character should be separated with line, as in other tables.
Comments on the Quality of English LanguageEditing of English language required.
Author Response
Attached the review response.

Reviewer 4 Report
Comments and Suggestions for Authors
I have read with interest the manuscript entitle “Prevalence of Multi-Drug Resistant and ESBL Producing Bacte-2 rial Pathogens in Patients with Chronic Wound Infections and 3 Spinal Cord Injury Admitted to a Tertiary Care Rehabilitation 4 Hospital”. In this manuscript, the authors assess in a retrospective single-center study the prevalence of different bacterial pathogens in patients with pressure ulcers admitted with spinal cord injury.
The structure of the manuscript and methodology used, including the statistical analysis is adequate to the hypothesis investigated.
The following few changes may improve the quality of the manuscript:
Abstract:
· Line 37-38: “Of the MDR bacterial isolates, 25.61% were methicillin-resistant Staphylococcus aureus, and 17.73% of extended-spectrum beta-lactamases”. Consider change to “extended-spectrum beta-lactamases Enterobacteriaceae”
Introduction
· Line 67: SA is not a common abbreviation of Staphylococcus aureus. Consider change to S. aureus (accepted abbreviation) after the first-time mention in the manuscript. Also, for the rest of microorganisms described, after the first mention, the standardized nomenclature is the same (ie; P. aeruginosa)
· Line 92-93: Dinh et al.: Include this reference as a number
· Page 3, lines 95-121; Consider summarizing this paragraph. It is not the main aim of the study to debate whether topical treatment is more or less suitable for this type of infections.
· Line 122: “In general, MDR organisms are not well understood,”. Consider change to “MDR organism isolations in soft tissue samples are not well understood”
Methods
· How was the sample size calculated?
· Line 169: Proportional odds logistic regression adjusted by age and sex. The title include the term “Low-Income Older Adults”. Therefore, the income level should be included in the adjusted analysis
Results
· Table 1. Baseline characteristics of patients. Consider include 2 decimals in all the frequency numbers.
· It would be interesting to include a Table or graph with the most prevalent drugs with anticholinergic effects prescribed in the population studied.
· Line 155-157: The criteria used to differentiate between colonization and infection was based on clinical assessment of the following signs or symptoms,….” This part should be included in “method” section
· Line 161 “Edsberg et al., 2016”. Include the reference with a number
· Line 229: MDR definition should be included in method section
· Table 4 is difficult to read. Consider other format or include abbreviation
Discussions
· These results support the findings by [1]. Consider change to “support the findings by Dinh, A. et al.” [1]
· An important limitation that should be considered is that the authors do not reflect the previous antibiotic pressure of the patients included. This point has a strong relationship with the type of bacterial isolated. This should be considered in the discussion,
Author Response
Attached the review response.

Round 2
Reviewer 1 Report
Comments and Suggestions for Authors
The authors followed the recommendations and the manuscript was substantially improved.